# OpenReview forum: "LongSafetyBench: Long-Context LLMs Struggle with Safety Issues"
_ICLR.cc/2025/Conference — ICLR 2025 Conference Withdrawn Submission_

### Official Review · Reviewer_aSHA · 2024-10-18

**Soundness:** 3
**Presentation:** 3
**Contribution:** 2
**Rating:** 3
**Confidence:** 4

**Summary:**

This paper introduces LongSafetyBench, which is specifically used to evaluate the safety of long context LLMs, filling an important gap in existing research field. During the implementation, this paper found that the proposed solution of using smaller safety-related datasets for supervision and fine-tuning can be simple and effective.

**Strengths:**

The paper introduces LongSafetyBench, a novel and valuable benchmark specifically designed to evaluate the safety of long-context LLMs. I appreciate the authors’ effort in addressing this important issue.

One key strength is the comprehensive set of experiments conducted on various advanced long-context LLMs, which solidifies the effectiveness of the proposed benchmark. The thorough evaluation of multiple models provides strong evidence of the benchmark’s utility. Additionally, during the benchmark-building process, the authors discovered that applying supervision fine-tuning with a small amount of data significantly enhances the safety of open-source LLMs in long-context scenarios. This finding is particularly valuable and has potential implications for improving the safety of models with minimal additional data.

**Weaknesses:**

The authors selected multiple metrics to evaluate safety performance in this paper. To maintain consistency throughout the experiments, it would be better to replace the score in Figure 3 with metrics that directly involve HA and SR. Additionally, some external evaluation methods and metrics, such as ManyShotJailbreak and SafetyBench, are not explained in sufficient detail, making the paper less self-contained and requiring readers to consult external resources for a full understanding of the evaluation framework.

The benchmark predominantly uses multiple-choice questions, which contrasts with traditional safety benchmarks designed for short-context scenarios that often involve more open-ended tasks. To better assess LLM safety in text generation, the benchmark should include more open-ended instructions. The mention of SR (line 207) may mislead readers into thinking that the benchmark involves text generation, which it does not.

Equating the model’s tendency to ignore harmful information with a weakening of safety seems questionable. The authors should consider designing more nuanced metrics to better reflect how safety performance varies with context length, as simply overlooking harmful content does not necessarily mean the model is unsafe.

Typos:
1. line 404: “memtioned” should be “mentioned”
2. line 537: “imporce” should be “improve”

**Questions:**

**Q1**: While your paper focuses on long-context safety issues, it doesn’t provide a detailed comparison with short-context LLMs or benchmarks, limiting the understanding of performance differences. Could you please add more insights about the reasons of safety variation from short context to long context?

**Q2**: Your benchmark heavily uses synthetic data or generated data, raising concerns about how well the results translate to real-world usage. Could you reconsider or explain how your benchmark works with real-world tasks?

**Q3**: While your paper shows that fine-tuning improves safety performance, has there been any analysis on whether this fine-tuning negatively impacts the model’s performance on other non-safety-related tasks? Are there any trade-offs between improving safety and maintaining general task performance, especially in long-context scenarios?

**Details Of Ethics Concerns:**

The authors needs to clarify whether their use of LLMs such as GPT-4 to synthesize harmful data violates the ethical policies of relevant service vendors.

---

### Official Review · Reviewer_grTc · 2024-10-20

**Soundness:** 2
**Presentation:** 2
**Contribution:** 2
**Rating:** 3
**Confidence:** 5

**Summary:**

This paper focuses on the long-context safety of large language models. The authors introduce the first benchmark for evaluating long-context safety, which uses multiple-choice questions. The dataset includes ten different tasks, with an average length of 41,889 words. Using this dataset, the authors tested ten existing advanced models. The results show that these models still need improvement in terms of long-context safety. Finally, the authors fine-tuned two models, LLaMA3-8b-instruct and InternLM2.5-7b-chat, using long-context safety data. They found that this fine-tuning significantly improved the safety of these models.

**Strengths:**

1. This paper introduces the first long-context safety benchmark.
2. The authors evaluate 10 popular models on this dataset and identify their insufficient safety performance.
3. The authors find that fine-tuning the model on automatically constructed long-context safety data could improve long-context safety.

**Weaknesses:**

1. This paper's judgment or definition of model behavior safety might be debatable and confusing. There are a few reasons for this:
-  (a) The model is actually doing multiple-choice questions, giving answers like A, B, C, D. Whether this counts as harmful behavior is not very clear, because the attacker doesn't get any harmful information they didn't already have.
-  (b) Some of the tasks defined in the model make me even more confused about point (a). For example, in the task "CountingCrimes," the question is "What crimes has Devil committed?" If the model honestly chooses the option that provides the answer, is this really unsafe?

2. The author emphasizes in the paper that "as the context length increases, models tend to exhibit more safety issues". However, they didn't actually test this claim on their own dataset. Specifically, the author didn't provide a comparison of the model's safety performance on LongSafetyBench data across different length ranges.

3. While the author shared some interesting findings, such as "while the model is capable of detecting harmful information, it tends to ignore such content," overall, the paper doesn't provide a deep analysis of the performance and reasons behind the safety of existing models in long context setting. Instead, it simply reports the evaluation results.
Suggestions for adding experimental analysis or discussion about:
- What is causing such noticeable safety issues with long context? Is it the length of the text, the multi-choice question format, or are there other factors?
- For failure cases with safety issues, have the model provide an explanation using a chain of thought or some other method. This could help provide more insight into how the model fails and how we could improve safety.


4. The authors did not provide the dataset, and they also didn't promise to make the dataset public in the future.

**Questions:**

1. Can you show results regarding how the model performs in different length ranges when tested using LongSafetyBench?
2. Can you provide one safe case and one unsafe case for each task?

---

### Official Review · Reviewer_gNDM · 2024-10-25

**Soundness:** 1
**Presentation:** 2
**Contribution:** 2
**Rating:** 3
**Confidence:** 4

**Summary:**

This paper presents a new benchmark for evaluating the safety of large language models (LLMs) under long-context scenarios. The authors construct 1,201 multiple-choice questions according to ten safety-related tasks and three harmfulness levels to assess the safety of long-context models from various perspectives. Extensive experiments are conducted with eight commonly used LLMs, and the experimental results indicate that existing models suffer safety issues in long-context scenarios compared with the shot-context counterparts. In addition, the authors notice that supervised fine-tuning (SFT) with a small portion of long-context data can greatly enhance the safety performance of state-of-the-art (SOTA) open-source models.

**Strengths:**

1. A new benchmark to assess the safety properties of LLMs under long-context scenarios.
2. Large-scale experiments, including eight LLMs on ten long-context safety tasks.

**Weaknesses:**

This paper mainly studies the safety of long-context LLMs to probe whether the long-text inputs can degrade the safety capabilities of such models. However, the construction of the benchmark and the evaluation of the model lack some technical details and corresponding justifications, which thereby makes the usability of the benchmark and the validity of the evaluation results questionable. Please see the detailed comment below.

1. My biggest concern is the design of the tasks lacks a systematic taxonomy and adequate justification. Namely, why select such ten types of tasks, and to what extent can they provide a comprehensive safety evaluation? Also, it is not clear how the fully harmful context and partially harmful context are identified, as both of them are formed with long-context inserted harmful information. To what degree can a question be considered fully harmful instead of partially harmful?

2. In addition to the previous point, the formats of some tasks (e.g., HarmfulExtraction, HarmfulTendency, ManyShotJailbreak) are quite similar. To me, it is not clear why they are classified as different tasks. Can each of them measure a distinct safety perspective of LLMs, or are they just similar tasks but different topics? This issue also applies to the tasks under the category of partially harmful context.

3. Instead of a free-form QA-style generation, the authors construct the questions following a multiple-choice scheme with four options representing different safety degrees. To me, this scheme is more like a measurement regarding the safety awareness of the context instead of the safety of the generation. Moreover, the options for each task are generated by Qwen2 according to "predefined option construction standards." What are the predefined option construction standards for each task?  Is there a validation process taken to verify the options generated by Qwen since such generated options can be objective?

4. In Section 3.3.3, the authors mention: "...presented with a seemingly safe and reasonable scenario...". What does the term "seemingly safe" apply to?

5. The evaluation metric safe response (SR) require more justification. To me, this metric only indicates to what extent the subject LLM can accurately select the safest option, but not the ability to generate a safe response w.r.t the given context and question.

6. In Section 4.2, the authors claim the safety performance of the subject models in long-context scenarios does not perform as well as in short-context cases. However, this claim is made by comparing the ranks of the models between the experiments from this paper and a third-party leadboard. This does not seem like a valid comparison as the evaluations are not conducted from the same scope, tasks and metrics. To justify the claim made by the authors, there should be some contrastive evaluation to assess the safety performance of models with the same question but under long-context and short-context scenarios.

7. The authors conduct SFT on two open-source models with data from four tasks and report that the performance of fine-tuned LLMs is comparable to SOTA closed-source models. I am concerned about the generability of this conclusion, as the same dataset is used for tuning and evaluation. Also, it seems the authors only use one closed-source LLM, Claude-3.5-sonnet, to draw a conclusion that is somehow not really convincing.

8. The presentation of this paper can be further improved as there are many grammar issues and ambiguous statements. For example, Table 2 and Figure 6 are not discussed in the paper, metrics (HA and SR) are mentioned at the beginning with no explanations, and the texts in the Figures, such as Figure 2, are hard to recognize.

9. I did not find any access to the benchmark or any replication material. Therefore, I cannot make any comment on the verifiability.

**Questions:**

Please see the detailed comments. I am especially looking forward to the author's response regarding comments 1, 2, 3 and 6.

---

### Official Review · Reviewer_bV9u · 2024-11-09

**Soundness:** 3
**Presentation:** 2
**Contribution:** 2
**Rating:** 5
**Confidence:** 5

**Summary:**

The paper introduces LongSafetyBench, a benchmark designed to objectively and comprehensively evaluate the safety of long context models. It consists of 1201 multiple choice questions covering 10 different tasks.

**Strengths:**

1. The paper attempts to evaluate the safety of long context models. There will be a lot of emphasis on long context LLMs. With long context safety concerns are exacerbated.
2. The paper finds that existing models exhibit insufficient safety capabilities

**Weaknesses:**

1. The corpus is not large enough to be a proper benchmark for evaluating safety capabilities of LLMs. One should rather attempt  benchmarks for particular domains (e.g., medical, political, etc.). That would help mitigate the drawbacks associated with the relatively smaller corpus. Also it would enable the authors to address biases in individual domains and keep the safety evaluation "balanced".
2. The authors have used crime novels and extracted criminal acts from them (HarmfulExtraction).  There was no attempt to distinguish between fact and fiction. Certain things possible in the fictional world may not be possible in the real world. Since the safety evaluation is being done for the real world, the fictional actions should not be part of the corpus. I would rather want to see a corpus created by interviewing law enforcement officers. Also a comparison between evaluation based on corpus created from crime novels and that obtained by interviewing detectives or law enforcement officers would be interesting. It will help us understand the difference  between the two distributions.
3.  Similarly, tasks such as HarmfulAdvice should be constructed in consultation with real world professionals such as medical professionals. The corpus should take into account the cultural or legal variations in different countries or regions and the contextual factors that influence the behavior of the population of a particular region (ethnic background, social conditions, etc.).

**Questions:**

1. Safety also depends on context. For example, laws of a particular country.  Have you considered that in creating your benchmark
2. How did you construct the benchmark of LeadingQuestion?
3. I think you need to consider human feedback regarding the safety quesyions

---

### Note · Authors · 2024-11-19

I have read and agree with the venue's withdrawal policy on behalf of myself and my co-authors.